# Interaction between GRP78 and IGFBP-3 Affects Tumourigenesis and Prognosis in Breast Cancer Patients

**DOI:** 10.3390/cancers12123821

**Published:** 2020-12-18

**Authors:** Hanna A. Zielinska, Carl S. Daly, Ahmad Alghamdi, Amit Bahl, Muhammed Sohail, Paul White, Sarah R. Dean, Jeff M. P. Holly, Claire M. Perks

**Affiliations:** 1IGFs & Metabolic Endocrinology Group, Bristol Medical School, Translational Health Sciences, University of Bristol, Southmead Hospital, Bristol BS10 5NB, UK; hanna.zielinska@hotmail.com (H.A.Z.); ahmad.alghamdi@bristol.ac.uk (A.A.); jeff.holly@bristol.ac.uk (J.M.P.H.); 2Faculty of Health Sciences, University of the West England, Bristol BS16 1QY, UK; dalycs@googlemail.com (C.S.D.); Paul.White@uwe.ac.uk (P.W.); sarah4.dean@uwe.ac.uk (S.R.D.); 3Faculty of Applied medical Sciences, Taif University, Taif, Saudi Arabia; 4Bristol Haematology and Oncology Centre, Department of Clinical Oncology, University Hospitals Bristol, Bristol BS2 8ED, UK; amit.bahl@uhbristol.nhs.uk; 5Faculty of Life Sciences, School of Cellular and Molecular Medicine, Bristol University, Bristol BS8 1TD, UK; m.sohail@bristol.ac.uk

**Keywords:** breast cancer prognosis, GRP78, IGFBP-3, metastasis

## Abstract

**Simple Summary:**

We investigated the clinical significance of a recently identified association between insulin-like growth factor binding protein 3 (IGFBP-3), that plays a key role in breast cancer progression, and glucose-regulated protein 78 (GRP78). We report a direct correlation between the expression of GRP78 and IGFBP-3 in breast cancer cell lines and tumour sections. Kaplan–Meier survival plots revealed that patients with low GRP78 expression that are positive for IGFBP-3 had poorer survival rates than those with low IGFBP-3 levels. With breast cancer cells we showed that knock-down of GRP78 negated the entry of IGFBP-3 into the cells and this switched the action of IGFBP-3 from promoting to inhibiting cell death. Together, our data suggest that loss of GRP78 reduces IGFBP-3 entry into cells switching its actions to promote tumorigenesis and predicts a poor prognosis in breast cancer patients.

**Abstract:**

Insulin-like growth factor binding protein 3 (IGFBP-3) plays a key role in breast cancer progression and was recently shown to bind to the chaperone protein glucose-regulated protein 78 (GRP78); however, the clinical significance of this association remains poorly investigated. Here we report a direct correlation between the expression of GRP78 and IGFBP-3 in breast cancer cell lines and tumour sections. Kaplan–Meier survival plots revealed that patients with low GRP78 expression that are positive for IGFBP-3 had poorer survival rates than those with low IGFBP-3 levels, and we observed a similar trend in the publicly available METABRIC gene expression database. With breast cancer cells, in vitro IGFBP-3 enhanced induced apoptosis, however when GRP78 expression was silenced the actions of IGFBP-3 were switched from increasing to inhibiting ceramide (C2)-induced cell death and promoted cell invasion. Using immunofluorescence and cell surface biotinylation, we showed that knock-down of GRP78 negated the entry of IGFBP-3 into the cells. Together, our clinical and experimental results suggest that loss of GRP78 reduces IGFBP-3 entry into cells switching its actions to promote tumorigenesis and predicts a poor prognosis in breast cancer patients.

## 1. Introduction

Breast cancer is the most common cancer in women globally and the second most prevalent cancer overall [1]. During the period from 2015 to 2017, Cancer Research UK reported 55,176 new cases of breast cancer with 11,399 deaths from the disease [2]. Localised disease is considered clinically manageable and death from breast cancer is generally due to the spread of the disease to other parts of the body. As many as 30% of patients with early breast cancer develop this type of disease: with the majority of these being resistant to current therapies [3].

The insulin-like growth (IGF) axis is composed of two ligands, IGF-I and -II, receptors and six high affinity IGFBPs (16–) and there is strong evidence to suggest that its dysregulation is associated with the risk and progression of many cancers, including breast cancer [4]. Insulin-like growth factor binding protein-3 (IGFBP-3) is the main IGF binding protein in human serum, where it transports and controls the bioavailability and half-life of the IGFs. Whilst it is mainly synthesised in the liver, it is also produced locally in both normal and tumour cells, including those of the breast [5]. Locally IGFBP-3 can exert both positive and negative effects on cell function depending upon the cellular environment [6]. With breast cancer cells, IGFBP-3 can promote cell survival of breast cancer cells when they were plated on fibronectin, that is representative of a more advanced cancer, but enhancing apoptosis when plated on collagen, that is representative of a more normal epithelial environment [7]. In tumours from patients with breast cancer, high levels of IGFBP-3 have been consistently associated with worse outcome and may therefore be a promising potential therapeutic target [8,9,10,11]. To understand the mechanism of action of IGFBP-3 studies have investigated novel IGFBP-3 binding partners. The protein GRP78 was recently identified in two studies [12,13]. The most traditional role for GRP78 is as a molecular chaperone for which it is found in the lumen of the endoplasmic reticulum. However, many studies have now shown that GRP78 can relocate to other sites of the cell including the plasma membrane implying that GRP78 may be involved in additional cellular functions [14]. 

There is very little data on the significance of the novel association reported to occur between IGFBP-3 and GRP78. The two initial publications reported opposite effects on cell function, promoting either cell survival or apoptosis. In this study we further investigated the functional implications and the clinical relevance of the association between these two proteins, using human breast cell lines and tissue from breast cancer patients. 

## 2. Results

### 2.1. Correlation between the Expression of GRP78 and IGFBP-3 in Breast Cell Lines and Clinical Samples

We examined the basal abundance of GRP78 and IGFBP-3 in nonmalignant MCF-10A cells and breast cancer cell lines using Western blotting. Among the five cell lines, GRP78 staining was strongest in the highly invasive and metastatic ERα-negative Hs578T and MDA-MB-231 cells, the nontumorigenic MCF-10A cells were intermediate and the noninvasive ERα-positive MCF-7 and T47D cells had the lowest levels of GRP78 (Figure 1(Ai,Aii)). Analysis of IGFBP-3 in this panel of cell lines revealed the highest levels of IGFBP-3 in the Hs578T cells, lower levels observed in MDA-MB-231 cells, with no detection of IGFBP-3 in the MCF-10A, MCF-7 and T47D cells (Figure 1(Ai, Aii)). We next confirmed previous in vitro data showing that GRP78 and IGFBP-3 associate by using IP experiments (Figure 1(Bi,Bii)). Additionally, we performed IHC on 69 formalin-fixed, paraffin-embedded breast cancer sections to assess levels and localisation of GRP78 and IGFBP-3. Representative examples of IHC staining to indicate weak, moderate, and strong staining are shown in Figure 1C. These results also suggest that the abundance of the two proteins are correlated (Figure 1D). Collectively, these results indicate an association between GRP78 and IGFBP-3.

### 2.2. The Absence of GRP78 Modifies the Actions of IGFBP-3 In Vitro

To evaluate the functional significance of the interaction between GRP78 and IGFBP-3 we manipulated the expression of GRP78 using siRNA and assessed changes in cell survival and invasion of Hs578T and MDA-MB-231 breast cancer cells treated with IGFBP-3 in the presence or absence of GRP78. We confirmed our previous observation that IGFBP-3 enhances apoptosis induced in Hs578T cells [15] with GRP78; IGFBP-3 increased ceramide (C2)-induced cell death (from 34% to 46%). However, with GRP78 silenced, the actions of IGFBP-3 were switched, with IGFBP-3 inhibiting C2-induced apoptosis (from 47% to 25%). (Figure 2(Ai,Aii)). With GRP78 silenced, IGFBP-3 acted as a survival factor. A switch in IGFBP-3 function upon GRP78 silencing was also shown in transwell invasion experiments. With Hs578T cells we observed a marked decrease in invasion upon treatment with IGFBP-3 and nonsilencing of siRNA, and this switched when GRP78 was knocked-down with IGFBP-3, then promoting invasion (Figure 2(Bi–Biii)). We obtained similar results using MDA-MB-231 cells (Figure 2(Ci–Civ)). These results suggest that without GRP78, IGFBP-3 acts to promote tumorigenesis.

### 2.3. IGFBP-3 Subcellular Localisation Is Regulated by GRP78

To clarify where IGFBP-3 exerted the different effects on the cells when GRP78 was silenced, we performed immunofluorescence analysis of Hs578T cells exposed to IGFBP-3 with and without GRP78 knocked-down. Figure 3A shows that adding exogenous IGFBP-3 increased levels of IGFBP-3 and GRP78 in the cells. Silencing GRP78 resulted in no endogenous or exogenous IGFBP-3 being observed in the cells. We obtained similar results using MDA-MB-231 cells (Figure 3B). In order to assess changes in the abundance of cell surface (cs) proteins, we undertook biotinylation experiments which indicated that exogenous addition of IGFBP-3 increased the abundance of and translocation of GRP78 to the cell surface (Figure 3(Ci,Cii)). These results suggest that without GRP78, IGFBP-3 cannot enter the cell.

### 2.4. Loss of GRP78 Predicts a Poor Prognosis in Breast Cancer Patients

We next evaluated the clinical relevance of our in vitro observations by examining the status of GRP78 and IGFBP-3 in tumour sections and how this related to breast cancer prognosis. Having demonstrated that loss of GRP78 promotes the tumorigenic actions of IGFBP-3 in our in vitro experiments we re-grouped our clinical dataset into GRP78 low tumours with either low or high IGFBP-3 staining. Although not statistically significant (*p* = 0.134), Kaplan–Meier survival analyses suggested that patients with low GRP78 staining that are positive for IGFBP-3 had poorer disease-free survival rates than those with low IGFBP-3 levels (Figure 4A). Representative examples of IHC staining are shown in Figure 4B. With this suggestive data from our small exploratory cohort, we then examined this relationship in the much larger publicly available METABRIC gene expression database. We found a significant (*p* < 0.003) trend towards poorer overall survival for patients with breast tumours with low GRP78 and high IGFBP-3 mRNA (Figure 4C). When GRP78-negative cases were compared according to IGFBP-3 gene expression profile the survival was found to be poorer among the GRP78-negative/IGFBP-3-positive subgroup of patients, which was consistent with our small exploratory clinical analyses and with our in vitro data. Collectively, these results highlight a role of GRP78 and IGFBP-3 in determining the outcome for breast cancer patients.

## 3. Discussion

Given a wealth of clinical and experimental evidence implicating a role for IGFBP-3 in the progression of breast cancer [10,11], IGFBP-3 represents a promising target for developing therapeutics for the clinical management of the disease. In searching for proteins associated with IGFBP-3, two recent reports identified the chaperone protein GRP78 as a novel binding partner of IGFBP-3, one using tandem affinity purification (TAP) and the other by yeast two-hybrid screening [12,13]. Whilst both groups showed an interaction between GRP78 and IGFBP-3 using MCF-7 breast cancer cells they reported opposite functional consequences: one group showed this interaction resulted in increased survival [12], whereas the other showed that it culminated in an increase in apoptosis [13]. While the functional implications of the association between the two proteins are being investigated, no studies have yet addressed the clinical relevance of their interaction.

Our analysis of basal levels of GRP78 and IGFBP-3 in different breast cancer cell lines revealed a similar pattern of abundance, with higher levels of both proteins in the more aggressive breast cancer cells compared with the less invasive cell lines. It should be noted that previous reports documenting an interaction between GRP78 and IGFBP-3 were mainly based on experimental models with overexpressed GRP78 [12,13]. We were able to demonstrate an association between the two proteins without modifying the basal levels of the proteins, suggesting the association to be physiologically relevant. 

We have shown previously that the ability of IGFBP-3 to modulate inducers of apoptosis is matrix-dependent, with exogenous IGFBP-3 promoting cell survival of breast cancer cells when they were plated on fibronectin but enhancing when exposed to collagen [15]. In our in vitro experiments we found that IGFBP-3 can also undergo a functional switch from being a tumour suppressor to a tumour promoter according to GRP78 status. On silencing GRP78 in breast cancer cells we observed a switch in the actions of IGFBP-3 from increasing to inhibiting apoptosis and inhibiting to promoting invasion. 

Available evidence suggests that GRP78 and IGFBP-3 can translocate between different cellular compartments [16,17,18]. Cell surface translocation of GRP78 has been described as a characteristic of cancer cells [19,20]. Li et al. found that cell surface GRP78 enhanced migration and invasion of colorectal cancer cells in a protease-dependent manner [21]. Zhang and colleagues identified cell surface GRP78 as the main inducer of epithelial to mesenchymal transition and promoter of cell invasion in hepatocellular cancer cells [22]. We suspect that subcellular localisation of IGFBP-3 holds the key to understanding changes in cell phenotypes that we observed when GRP78 was silenced. Our data suggest that GRP78 controls IGFBP-3 localisation, depending on the presence or absence of GRP78; IGFBP-3 either enters the cell or stays on the cell surface. Using immunofluorescence and cell surface biotinylation, we showed that GRP78 aids translocation of IGFBP-3 into the cell and that this process is impaired when GRP78 is knocked-down. Without GRP78, IGFBP-3 cannot enter the cell, and this shifts IGFBP-3 from being a promoter of apoptosis to a survival factor and from an inhibitor to a promoter of invasion. 

From these experiments we cannot say definitively whether GRP78 binds to folded or unfolded IGFBP-3 or both, but we have shown that GRP78 is required for uptake into the cell and as far as we are aware, it has not been established whether the actions of IGFBP-3 within the cell and the nucleus are mediated by folded or unfolded IGFBP-3. However, the recombinant IGFBP-3 that we add is predominantly folded (and as we show is biologically active) and this is taken into the cell with GRP78, but not without GRP78. It is also generally considered that cell secreted IGFBP-3 is folded, so our data again would support GRP78 binding to folded IGFBP-3, but further work would be needed to confirm this.

In a cell model we previously reported that IGFBP-3 could bind to the β-1-integrin and caveolin-1 and enhance or inhibit the association between these proteins in a matrix-dependent fashion [7]. GRP78 has also been linked to integrin receptor signalling on the cell surface of cancer cells [21]. These findings may suggest that GRP78 influences the degree of IGFBP-3 uptake into the cell in this manner via an interaction with integrins. Notably, these data indicate that in the absence of GRP78, IGFBP-3 is prevented from entering the cells and this restricts IGFBP-3 to the outside of the cell, where it can interact with cell surface molecules, such as caveolin 1 and integrin receptors [7]. It appears that when IGFBP-3 is unable to enter the cell it promotes survival and invasion. 

However, the presence of GRP78 enables IGFBP-3 to enter the cells, where it has opposite actions in promoting apoptosis and inhibiting invasion, perhaps via its capacity to translocate to the nucleus [18]. In order to further explore the role of GRP78 in the actions of IGFBP-3 it will be interesting to overexpress GRP78 to clarify how IGFBP-3 activity may be affected in situation of endoplasmic reticulum stress.

The association between IGFBP-3 and GRP78 that we observed in vitro was further supported by our clinical study in which we identified a positive correlation between GRP78 and IGFBP-3 in tumor samples from patients with breast cancers. There has been no consensus over whether IGFBP-3 positively or negatively impacts breast cancer prognosis. The results are conflicting, with some studies correlating high tumour IGFBP-3 levels with worse clinical outcome [23,24,25], and others reporting the opposite [26,27]. The complexity of IGFBP-3 actions on breast cancer cells has long been acknowledged with reports of opposite effects and with the surrounding extracellular matrix microenvironment [15] and changes in the sphingolipid rheostat [28] having been described as underlying these different actions. Our data identifies GRP78 as a further important contributing factor. Through a combination of in vitro experiments, clinical analyses and assessment of data from the publicly available METABRIC gene expression database, we have provided evidence that the interaction between GRP78 and IGFBP-3 plays major role in breast cancer progression and prognosis. Results using the three approaches cumulatively suggests that loss of GRP78 promotes the tumorigenic actions of IGFBP-3 and predicts a poor prognosis in patients with breast cancer. Specifically, our analysis of patient specimens implied that overall survival was worse for those patients whose tumours are GRP78-negative/IGFBP-3-positive than for those with GRP78-negative/IGFBP-3 negative tumours. Consistent with this, an assessment of IGFBP-3 mRNA levels in GRP78 mRNA negative cases from METABRIC database also revealed a trend towards poorer overall survival and disease-free survival in breast tumours with low GRP78 and high IGFBP-3 mRNA. 

The discovery that GRP78 can determine whether IGFBP-3 is pro or anti-tumorigenic paves a new way for clinical prognosis. Clinically, high GRP78 levels have been associated with more aggressive features and worse prognosis in various cancers in many studies [29,30,31,32,33] however, contradictory results have also been reported [34,35,36]. Our results indicating that the function of IGFBP-3 is either pro- or anti-tumorigenic depending on levels of GRP78, suggest that the key to improving the clinical outcome of patients is to implement measurement of both markers in breast tumours in clinical practice.

## 4. Materials and Methods

### 4.1. Cell Culture

Human breast cancer cell lines MCF-7, T47D, MDA-MB-231 and Hs578T were purchased from ATCC (Teddington, Middlesex, UK) and the MCF10A, a nontumorigenic epithelial cell line was purchased from the American Type Culture Collection (ATCC, Manassas, VA, USA). These cells have been authenticated by short tandem repeat (STR) analysis and had been confirmed as mycoplasma-negative in our routine quality control. All cell lines were cultured as described previously [28,37,38].

### 4.2. Dosing with IGFBP-3 and siRNA Transfection

Human nonglycosylated IGFBP-3 was purchased from Insmed (Bridgewater Township, NJ, USA). Predesigned gene-specific siRNAs for GRP78 (HS_HSPA_2; target sequence 5′-AACTGTTACAATCAAGGTCTA-3′), and nonsilencing siRNA (AllStars negative control) were bought from Qiagen (20 nM; Benelux, Venlo, The Netherlands). GRP78 silencing was also performed with a second source of siRNA: ON-TARGET plus Human HSPA5 (3309) siRNA-SMARTPOOL (Target Sequences: GCGCAUUGAUACUAGAAAU; GAACCAUCCCGUGGCAUAA; GAAAGAAGGUUACCCAUGC; AGAUGAAGCUGUAGCGUAU) Downregulation of GRP78 was achieved via transfection using Saint-red (Synvolux, Leiden, The Netherlands) according to the manufacturer’s protocol.

### 4.3. Transwell Invasion Assay

Control and siRNA transfected cells (1.5 × 10), with and without treatment with IGFBP-3 (100 ng/mL), were plated in 500 µL of serum free media and added in triplicate to collagen-coated 8 μm pore-transwell inserts (Millipore, Watford, UK). DMEM containing 5% fetal bovine serum, used as a chemoattractant solution was added to the lower chamber. Cells were allowed to migrate toward the chemoattractant media in the lower compartment for 4  h at 37  °C. The migrated cells in the filter were fixed using 4% paraformaldehyde, permeabilised with triton X and stained with Crystal violet. The stain then dissolved in 1% SDS on plate shaker for one hour, before reading the optical density values using iMark plate reader (BioRad, Watford, UK) at 595 nm.

### 4.4. Western Blotting

Western blot analysis was performed as described previously [37]. Briefly, 30 μg of protein were run on 10% SDS-PAGE, transferred to nitrocellulose membrane (BioRad) and immunoblotted with the following antibodies: GRP78 (1:1000 BD Bioscience, San Jose, CA, USA), IGFBP-3 (1:1000 Santa Cruz, Dallas, TX, USA) and α Tubulin (1:5000 Merck Millipore, Burlington, MA, USA). After incubation with specific secondary antibodies conjugated to peroxidase (Sigma, St. Louis, MO, USA), proteins were visualised by Clarity ECL substrate (BioRad) using BioRad Chemidoc XRS + system and analysed using Image lab software (BioRad). 

### 4.5. Immunoprecipitation

Immunoprecipitation was performed as described previously [7]. In brief, 500 ug of cell lysate were incubated overnight with 2 μg of GRP78 antibody at 4 °C. Anti-mouse IgG (2 μL) were added and incubated for 1 h at 4 °C. Protein A/G PLUS-Agarose (25 μL) were added and incubated for 90 min at 4 °C. The immunoprecipitated sample was then collected by centrifugation at 3500 rpm for 3.5 min at 4 °C and supernatant was carefully aspirated and discarded. The pellet was then washed with lysis buffer (5×), the supernatant aspirated and the pellet was resuspended in 35 μL of 1× electrophoresis sample buffer in preparation for Western blotting.

### 4.6. Immunofluorescence Staining and Confocal Imaging

Immunofluorescence and confocal imaging were performed as described previously [7]. Control or siRNA transfected MDA-MB-231 and HS578T cells were seeded onto coverslips in 12-well plates using DMEM containing 5% fetal bovine serum and incubated for 24 h, then the media was replaced with serum free media for a further 24 h, before dosing with 100 ng/mL IGFBP-3 and incubating for 48 h. Cells on the coverslip then were fixed in 4% paraformaldehyde, permeabilised with 0.3% Triton X-100 for 20 min and blocked with 5% normal goat serum (Vector Laboratories, Burlingame, CA, USA) for 1 h at room temperature. The coverslips were incubated with primary antibody to GRP78 1:250 (BD Bioscience) and IGFBP-3 1:250 (Santa Cruz) at 4 °C overnight. They were then incubated with a goat anti-mouse secondary antibody (Thermo Fisher Scientific (Alexa Fluor 594); 1:400, Loughborough, UK) and a goat anti-rabbit secondary antibody (Thermo Fisher Scientific (Alexa Fluor 488); 1:400) respectively for 1 h at room temperature in 1% normal goat serum/PBS buffer. The coverslips were mounted with 4′,6-diamidino-2-phenylindole, dihydrochloride DAPI (Vector Laboratories, Burlingame, CA, USA) to stain the nuclei. Immunofluorescence images were captured using a Leica SP5 confocal laser scanning microscopy at the Wolfson Bioimaging facility, University of Bristol.

### 4.7. Cell Surface Biotinylation and Avidin Pulldown

Cells were washed with cold PBS (×3). EZ-Link Sulfo-NHS-SS-Biotin (Thermo Scientific, Waltham, MA, USA) in PBS at 110 mg/mL was added, and the cells were gently shaken for 30 min at 4 °C. The solution was removed, and the cells were rinsed 3 times with quenching buffer containing Tris-Cl, pH 7.4, in cold PBS to stop the reaction. Cells were lysed on ice using RIPA buffer. Cell lysate (500 μg) were incubated at 4 °C for 2 h with High Capacity NeutrAvidin Agarose Resin (Thermo Scientific) to purify the surface protein. The purified surface proteins were then collected by centrifugation at 3500 rpm for 3.5 min at 4 °C. The centrifugation cycle was then repeated 5× before removing the supernatant and resuspending the pellet in 35 μL of 2× Laemmli buffer and subjected to Western blotting.

### 4.8. Human Tissue Samples

Breast cancer tissue samples were removed surgically at Bristol Royal Infirmary under ethical approval from NHS Health Research Authority and UWE Ethics Committee (Ref. 11/SW/0127). The patients were informed and consented for using these samples in the study. Methods used were in accordance with the NHS Health Research Authority guidelines and regulations. The samples used were surplus to diagnostic requirements. The cohort was enriched for triple negative breast cancer due to initial hypotheses and previous work on these tissues (not published) but was selected to include samples of the main immunohistochemical subtypes (ER/PR/HER2/TN).

Table 1 provides the patient ages and clinicopathological characteristics of the tumour tissue from 69 participants.

### 4.9. Immunohistochemistry

Formalin-fixed, paraffin-embedded (FFPE) tissue (69) were serially sectioned at 4 µm using a microtome (Leica RM2235) and mounted on slides (Thermo Fisher Scientific, Loughborough, UK). Tissue sections were deparaffinised in histoclear (National Diagnostics, Atlanta, GA, USA) and rehydrated using a serial dilutions of ethanol concentrations and dH2O. Slides were then covered with 3% *v*/*v* hydrogen peroxide to quench endogenous peroxidase for 10 min at room temperature. Sections were then placed in 10 mM citrate buffer pH 6.0, boiled at 95 °C for 30 min using a water bath for antigen retrieval, followed by cooling to room temperature. 5% horse serum in tris-buffered saline (TBS) (200 mM sodium chloride, 2 mM tris, pH 7.5) was then used to block nonspecific binding sites for 1 h at room temperature. Sections were incubated with primary antibody (1:100 IGFBP-3 and 1:400 GRP78) in blocking solution. Staining assessment was determined based on two variables, the grade for intensity of staining (1, weak; 2, moderate; 3, strong) and percentage of stained cells (1, 0 to <10%; 2, 10 to <50%; 3, 50–100%). The overall scoring of IGFBP-3 and GRP78 expression was considered positive when both scores were ≥2 as described previously by [39,40].

### 4.10. Molecular Taxonomy of Breast Cancer International Consortium (METABRIC) Dataset Analysis

Z-scores for GRP78 mRNA were extracted from METABRIC (METABRIC, Nature 2012 & Nat Commun 2016) via the cBioPortal database. All breast cancer cases with an mRNA z-score of –2 or less were considered [40] low expression for GRP78; this resulted in 83 cases for subsequent analysis. Within this group IGFBP3 mRNA expression levels were correlated with overall survival. Disease-free survival information was not available. Cases with IGFBP3 z scores that were positive were considered as to have higher than mean expression levels and cases with negative z-scores were considered lower than mean IGFBP3 expression levels.

### 4.11. Statistical Analysis

Unless otherwise stated, all experiments were repeated three times, and each in triplicate. The data were analysed using GraphPad Prism 8.0.1 software for Windows (La Jolla, CA, USA), one-way ANOVA followed by least significant difference (LSD) post-hoc test. SPSS 24.0.1 software for windows was also used for the analyses of IHC data using Spearman’s rank correlation test, Chi square test, Fisher exact probability test and Kaplan–Meier survival test. In all analyses a *p* value of <0.05 was considered significant.

## 5. Conclusions

In conclusion, our study adds to our understanding of the cellular significance of the association between GRP78 and IGFBP-3 and of its clinical significance. We have shown that GRP78 is an important component in determining the effects of IGFBP-3 on cell phenotypes and in the survival of patients with breast cancer. Our study identified GRP78 and IGFBP-3 as useful biomarkers that might influence treatment decisions and provide new therapeutic targets.

## Figures and Tables

**Figure 1 cancers-12-03821-f001:**
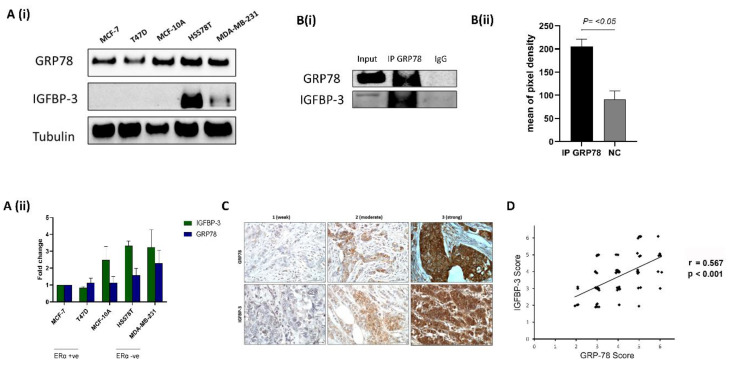
GRP78 and IGFBP-3 co-operate (**A,i**) The levels of GRP78 and IGFBP-3 were examined using Western blotting of the whole cell lysates from the nonmalignant MCF-10A cells and various breast cancer cell lines. (**A,ii**) Quantification of Western blot results shown in (**A,i**). (**B,i**) Hs578T cell extract was subjected to IP with either control IgG or GRP78 antibodies and the samples immunoblotted with GRP78 and IGFBP-3 antibodies. Input represents whole cell lysate as a percentage of protein amount (12.5%) used for the IP reaction. (**B,ii**) Quantification of IP results presented in (**B,i**). (**C**) Representative images of GRP78 and IGFBP-3 IHC staining (1 = weak; 2 = moderate; 3 = strong) in paraffin-embedded breast cancer tissue sections (*n* = 69). IHC scores of GRP78 and IGFBP-3 abundance are shown in (**D**) Nonparameter Spearman correlation co-efficiency and the p value are also shown. The whole blots (uncropped blots) are shown in the Appendix A.

**Figure 2 cancers-12-03821-f002:**
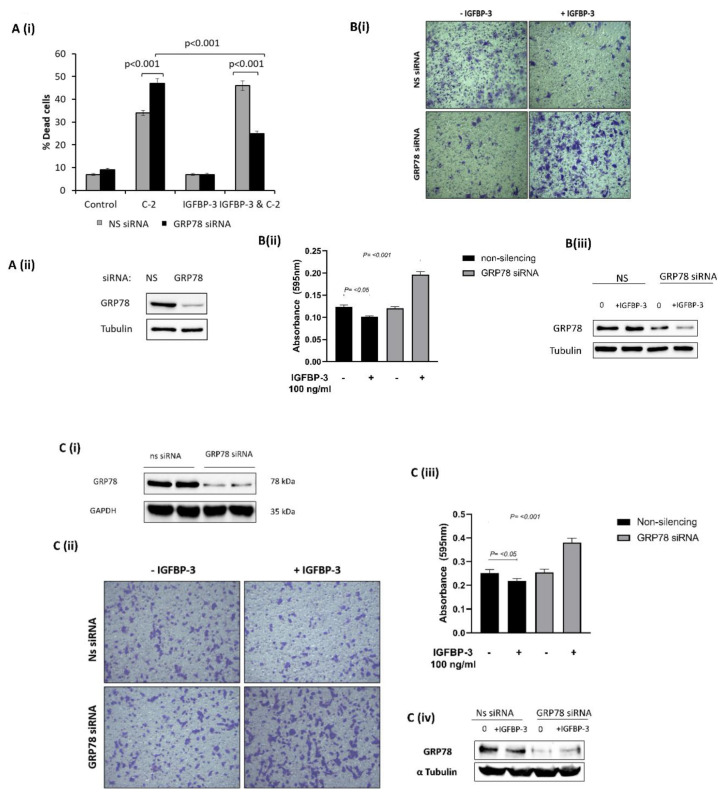
Loss of GRP78 alters the actions of IGFBP-3 on breast cancer cells. (**A,i**) The percentage cell death was assessed by trypan blue dye exclusion method in Hs578T cells treated with IGFBP-3 in combination with an apoptotic dose of C2 in the presence or absence of target siRNA to GRP78 or nonsilencing siRNA. Effective GRP78 silencing is shown in (**A,ii**). (**B,i**) Cell invasion was measured using the transwell invasion assay with crystal violet staining in Hs578T cells treated with IGFBP-3 with and without GRP78 knocked-down (×20). (**B,ii**) Quantification of stained invaded cells using a microplate reader. Effective GRP78 silencing is shown in (**B,iii**). (**C,i**) Representative Western immunoblot indicating effective silencing of GRP78 using siRNA (ON-TARGET plus Human HSPA5 (3309) siRNA-SMARTPOOL) (**C,ii**) Cell invasion was measured using the transwell invasion assay with crystal violet staining in MDA-MB-231 cells treated with IGFBP-3 with and without GRP78 knocked-down (×20). (**C,iii**) Quantification of stained invaded cells using a microplate reader. Effective GRP78 silencing is shown in (**C,iv**). The whole blots (uncropped blots) are shown in the Appendix A.

**Figure 3 cancers-12-03821-f003:**
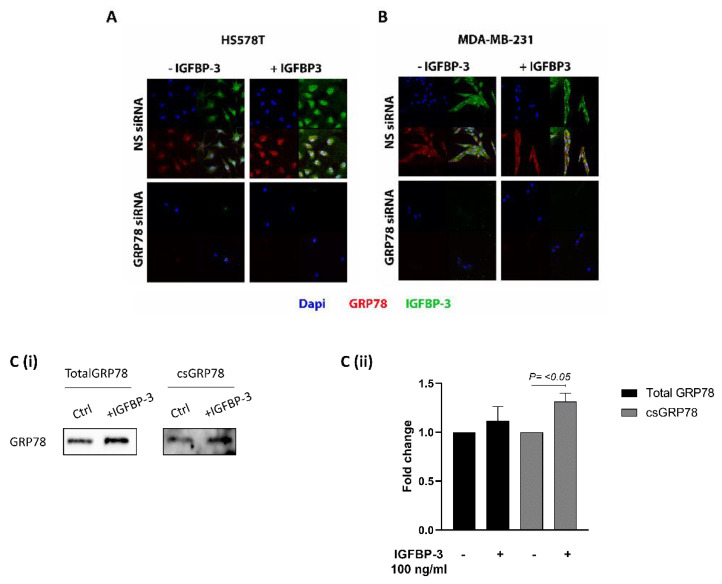
Knock-down of GRP78 blocks the entry of IGFBP-3 into the cells. IF staining of GRP78 and IGFBP-3 in cells treated with IGFBP-3, in the presence or absence of target siRNA to GRP78, or NS siRNA in (**A**) HS578T and (**B**) MDA-MB-231 cells (Original magnification 63×). (**C,i**) Hs578T cells treated with IGFBP-3 with and without GRP78 knocked-down were subjected to cell surface (cs) biotinylation followed by Western blot analysis. (**C,ii**) Quantification of Western blot results shown in (**C,i**). The whole blots (uncropped blots) are shown in the Appendix A.

**Figure 4 cancers-12-03821-f004:**
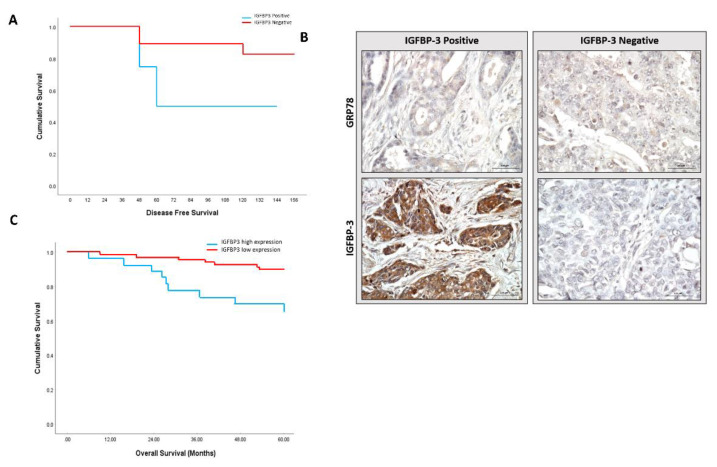
Loss of GRP78 is associated with worse clinical outcome. (**A**) Kaplan–Meier plots of disease-free survival for low and high staining of IGFBP-3 in GRP78-negative tumours (*n* = 26; *p* = 0.134). (**B**) Representative images of GRP78 and IGFBP-3 IHC staining. (**C**) The overall survival of breast cancer patients was analysed in the METABRIC database in GRP78-negative tumours based on IGFBP-3 gene expression profile (*n* = 83; *p* = 0.003). Statistically significant differences were determined by two-sided log-rank test.

**Table 1 cancers-12-03821-t001:** indicates patient ages and the clinicopathological characteristics of the tumour tissue from 69 participants with breast cancer (TN = triple negative breast cancers).

		Total
Age	<50	11
	50–70	41
	>70	17
Tumour size	≤2 cm	34
	2 cm–≤ 5 cm	33
	>5 cm	2
Grade	1	2
	2	30
	3	37
ER	Pos	23
	Neg	46
PR	Pos	18
	Neg	51
Her2+	Pos	25
	Neg	44
TN	Yes	30
	no	39
Histology	Ductal	59
	other	10
Lymph involvment	Yes	25
	No	44

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
