# Peer review of "Interaction between GRP78 and IGFBP-3 Affects Tumourigenesis and Prognosis in Breast Cancer Patients"

_cancers, 2020, doi:10.3390/cancers12123821_

Round 1
Reviewer 1 Report
The authors well addressed to reviewer's comments.
Author Response
Thank you very much for your suggestion.
Reviewer 2 Report
The revised manuscript improved significantly.
Author Response
Thank you very much for your suggestion
Reviewer 3 Report
The manuscript provided several lines of evidence that the interaction between GRP78 and IGFBP-3 affects breast cancer progression. In addition, loss of GRP78 diminished IGFBP-3 entry into cells, resulting in up-regulating tumorigenesis. The presented data suggested that interaction between GRP78 and IGFBP-3 plays a role in tumorigenesis and prognosis of breast cancers.
Major concerns:
- Hs578T cells express the high levels of IGFBP-3. However, IGFBP-3 is a secretary protein and the nascent IGFBP-3 is targeted into the lumen of ER. GRP78 is also present in the ER. GRP78 can associate with the misfolded or unfolded proteins. Can GRP78 associate with the unfolded and folded IGFBP-3? Which form of IGFBP-3 was pulled down in Fig. 1B(i)? GRP78 can be translocated to the cell surface. Can GRP78 bring IGFBP-3 to the plasma membrane of Hs578T or MDA-MB-231 cells?
- Fig. 1B(i) should mark that the percentages of IP were used for Input.
- Hs578T and MDA-MB-231 cells contain endogenous IGFBP-3. However, in the panels of GRP78-knockdowned Hs578T (Fig. 3A) and MDA-MB-231 cells (Fig. 3B), no IGFBP-3 was present in the cells. Why? The image resolution of Fig. 3A and 3B was poor. Both Figures cannot support the conclusion that addition of exogenous IGFBP-3 increase the levels of complex on the plasma membrane.
- Fig. 3C(i) lacks the internal control of membrane protein.
- No statistically significant Kaplan-Meier survival analyses of disease-free survival for low and high staining of IGFBP-3 in GRP78-negative tumors were obtained. Fig. 5A cannot support the conclusion of this study.
Minor concern:
- The name of the assigned journal should be shown in Reference 15.
- The image resolution of Fig. 5A and 5C was poor.
Author Response
We thank the reviewer for the time they have taken to review our manuscript and for raising some important questions. We hope we have addressed these below:
Major concerns:
- Hs578T cells express the high levels of IGFBP-3.
However, IGFBP-3 is a secretary protein, and the nascent IGFBP-3 is targeted into the lumen of ER. GRP78 is also present in the ER. GRP78 can associate with the misfolded or unfolded proteins. Can GRP78 associate with the unfolded and folded IGFBP-3?
Which form of IGFBP-3 was pulled down in Fig. 1B(i)? GRP78 can be translocated to the cell surface. Can GRP78 bring IGFBP-3 to the plasma membrane of Hs578T or MDA-MB-231 cells?
We thank the reviewer for this interesting question.
From these experiments we cannot say definitively whether GRP78 binds to folded or unfolded IGFBP-3 or both: but we have shown that GRP78 is required for uptake into the cell and as far as we are aware, it has not been established whether the actions of IGFBP-3 within the cell and the nucleus are mediated by folded or unfolded IGFBP-3.
However, the recombinant IGFBP-3 that we add is predominantly folded (and as we show is biologically active) and we show that it is taken into the cell with GRP78, but not without GRP78. It is also generally considered that cell secreted IGFBP-3 is folded, so our data again would support GRP78 binding to folded IGFBP-3. However, we have no data to indicate whether it does, or does not, bind to unfolded IGFBP-3: but as an ER-chaperone it should do, as that is its normal role.
For clarification of this interesting point we have included this explanation in the discussion (lines 189-195).
- Fig. 1B(i) should mark that the percentages of IP were used for Input.
We have indicated the percentage of IP used as input (12.5%) in the figure legend.
- Hs578T and MDA-MB-231 cells contain endogenous IGFBP-3. However, in the panels of GRP78-knockdowned Hs578T (Fig. 3A) and MDA-MB-231 cells (Fig. 3B), no IGFBP-3 was present in the cells. Why?
The reviewer quite rightly mentioned above that IGFBP-3 is a secreted protein that is then taken back into the cell. We conclude that in panels 3A & B with GRP78 silenced, the normal process of re-uptake of the secreted IGFBP-3 into the cells as observed in the controls has been inhibited and hence there is no IGFBP-3 in the cells. IGFBP-3 is secreted via the constitutive pathway and there are no intracellular stores and therefore if it does not re-enter via uptake then the endogenous levels would be expected to be very low and presumably below the sensitivity of our immunocytochemistry.
The image resolution of Fig. 3A and 3B was poor. We have resaved higher resolution versions and reinserted the panels to rectify this.
Both Figures cannot support the conclusion that addition of exogenous IGFBP-3 increase the levels of complex on the plasma membrane.
We have determined from these data that without GRP78, there is more IGFBP-3 on the cell surface and less inside the cells. We have not referred to changes in levels of complexes on the cell surface as we cannot make any conclusions pertaining to this from these data.
- Fig. 3C(i) lacks the internal control of membrane protein.
The experiment shown in Fig3Ci was conducted to examine proteins on the cell surface. We performed this by biotin labelling all the proteins on the surface of the cell and then precipitating them using agarose resin. We ran all the cell surface protein lysate on a gel and probed with GRP78 to determine if the levels of this protein on the cell surface varied in response to IGFBP-3. As the whole extract from each condition was loaded, they are compared to each other.
We also assessed changes in levels of GRP78 in response to IGFBP-3 in separate whole cell lysates.
Overall, these data indicate that GRP78 levels increased in the lysate but that in response to IGFBP-3 this increase was also observed specifically on the cell surface.
- No statistically significant Kaplan-Meier survival analyses of disease-free survival for low and high staining of IGFBP-3 in GRP78-negative tumors were obtained. Fig. 5A cannot support the conclusion of this study.
The reviewer is correct in that the data shown in Fig4a is not statistically significant (p=0.134), the Kaplan-Meier survival analyses showed a trend that patients with low GRP78 staining that are positive for IGFBP-3 had poorer disease-free survival rates than those with low IGFBP-3 levels.
We acknowledged that the cohort used to generate these data is small and hence the results are not statistically significant. However, the result we observed was confirmed in the much larger publicly available METABRIC gene expression database in which we found a significant (P<0.003) trend towards poorer overall survival for patients with breast tumours with low GRP78 and high IGFBP-3 mRNA which then supported our conclusions.
We feel that showing both sets of analyses is justified and we qualify this by indicating in the results that ‘collectively these results highlight a role of GRP78 and IGFBP-3 in determining the outcome for breast cancer patients’. The trend shown in the small cohort may not be statistically significant due to its size, but the trend may still indicate biological significance.
Minor concern: thank you for raising these points, which we have addressed below.
- The name of the assigned journal should be shown in Reference 15.
Reference 15 seems fine but on checking the refs again I did notice that the journal title was missing from ref 14- thank you- this has now been added.
- The image resolution of Fig. 5A and 5C was poor.
The figures have been re-saved and re-inserted to rectify the resolution issue.
This manuscript is a resubmission of an earlier submission. The following is a list of the peer review reports and author responses from that submission.
Round 1
Reviewer 1 Report
This study discovered that the interaction between GRP78 and IGFBP3 affects cancer progression in breast cancer patients. The molecular mechanism causing this consequence should be explored in this study.
Author Response
We thank the reviewer for taking the time to read our manuscript and for making such constructive comments. We hope we have addressed these below and have highlighted any changes in the text of the paper in yellow.
Reviewer 1: This study discovered that the interaction between GRP78 and IGFBP3 affects cancer progression in breast cancer patients. The molecular mechanism causing this consequence should be explored in this study.
We and others have been studying the mechanism of action of the intrinsic effects of IGFBP-3 in breast cancer cells for many years. I have provided a few links to our key papers.
https://www.ncbi.nlm.nih.gov/pmc/articles/PMC3356103/
https://www.ncbi.nlm.nih.gov/pmc/articles/PMC2998100/
https://academic.oup.com/endo/article/147/7/3484/2501228
The recent identification of a novel binding partner for IGFBP-3, prompted us to determine if GRP78 did associate with IGFBP-3 (in a model that did not involve artificial over-expression of either of the proteins) and if GRP78 influenced the intrinsic effects of IGFBP-3 in our established models. Furthermore, we assessed if this association, which did affect the IGF-independent effects of IGFBP-3, was relevant in actual clinical samples. Having established that GRP78 appeared to play a key role in the actions of IGFBP-3 by modifying its entry into the cells, this will be the basis of our future mechanistic work, but this was beyond the scope of this manuscript which already contains substantial new data.

Reviewer 2 Report
In this manuscript the authors studied the the functional implications and the clinical relevance of the association between IGFBP-3 with GRP78 in breast cancer.
Although interesting and with scientific content, this manuscript cannot be accepted in this way for the reasons indicated below:
-The introduction of the manuscript is weak and does not adequately contextualize the interest in carrying out a research work subject to the proposed theme.
-In my opinion both cell cultures and use of human tissues are included in in vitro studies. So, I disagree with the comment written in line 42.
-The authors do not explain the criteria used to include samples taken from women in this study.
-In this work, two types of breast cancer are described by the authors: Dcutal (n=59) and other (n=10). What are the others? How behaved the expression of IGFBP-3 with GRP78 in ductal tumours?
-None of the histological images has magnification as well as the images of cell culture.
-Figures with the histopathological pattern should be added.
- The references should be checked.
Author Response
We thank all the reviewer for taking the time to read our manuscript and for making such constructive comments. We hope we have addressed these below and have highlighted any changes in the text of the paper in yellow.
Reviewer 2: The introduction of the manuscript is weak and does not adequately contextualize the interest in carrying out a research work subject to the proposed theme.
We have substantially expanded the introduction to better explain the context and prepare the reader for the proposed study (lines 34-58)
In my opinion both cell cultures and use of human tissues are included in vitro studies. So, I disagree with the comment written in now line 58. To address this point, we have amended this sentence to read ‘using human breast cell lines and tissue from breast cancer patients’ (now line 58)
The authors do not explain the criteria used to include samples taken from women in this study. Samples used were surplus to diagnostic requirements, this has been added to the methods. The cohort is used for ongoing studies related to this topic area. We have also added the following information to the methods section: The cohort was enriched for triple negative breast cancer due to initial hypotheses and previous work on these tissues (not published) but was selected to include samples of the main immunohistochemical subtypes (ER/PR/HER2/TN) (now lines 288-291).
In this work, two types of breast cancer are described by the authors: Ductal (n=59) and other (n=10). What are the others? How behaved the expression of IGFBP-3 with GRP78 in ductal tumours? Others include lobular and mixed- this subtyping information was obtained from the histological reports. Due to the small cohort numbers, it is not possible to compare ductal tumour results to the other. This information was included to give readers information on the cohort.
None of the histological images has magnification as well as the images of cell culture.
We apologise for this oversight. All magnifications have been added to the figure legends and the histological images now all have scale bars.
Figures with the histopathological pattern should be added. We thank the reviewer for this suggestion, but we are not sure if this addition would be of benefit, as the majority of the tumours are of ductal type and we did not draw any comparisons between the other histological subtypes as the numbers were so small.
The references should be checked. All the references have been checked.

Reviewer 3 Report
This manuscript reported the cellular significance of the association between GRP78 and IGFBP-3 in breast cancer cells. This study identified that GRP78 is an important molecule in determining the effects of IGFBP-3.
Major points:
- The introduction is very short.
- It is hard to follow the results section. Please try to explain the results in detail for the readers to follow the content easily.
- Lines 51-52 “Analysis of IGFBP-3 in this panel of cell lines revealed a similar level of abundance”
When comparing the expression levels of Hs578T and IGFBP-3, there is a major difference in the expression levels in different cell lines. IGFBP-3 is completely absent in MCF-7, T47D and MCF-10A. The authors has to try to explain the observation from the figures.
- Better to show the immunohistochemistry data after the Western blot data, which will allow the readers to easily follow the content.
- The figure 3B i and ii is confusing. What is “csGRP78”. Explain clearly in the text.
- Figure 4 legends are wrong “(A) Representative images of GRP78 and IGFBP-3 IHC staining. (B) Kaplan-Meier plots of disease-free survival for low and high staining of IGFBP-3 in GRP78-negative tumors “. But Figure 4 indicates (A) Kaplan-Meier plots and (B) IHC staining.
- Please include all the supplementary figures in the result section.
- Include all the experiment results of MDA-MB-231.
- The authors have to consider experimenting with the overexpression of GRP78 in MCF-7/T47D.
- Any in vivo data is available to support the in vitro data?
Minor points
- Lines 48-49: “Among the five cell lines, GRP78 staining was strongest in the highly invasive and metastatic ERα-negative Hs578T and MDA-MB-231 cells”
The figure 1A i & ii indicates the protein expression using a Western blot experiment.
Please change the sentence to “Among the five cell lines, GRP78 expression level was strongest in the highly invasive and metastatic ERα-negative Hs578T and MDA-MB-231 cells”
- Lines 206-207 “Our results that the function of IGFBP-3 is either 207 pro- or anti-tumorigenic ….
Correct the sentence.
Author Response
We thank the reviewer for taking the time to read our manuscript and for making such constructive comments. We hope we have addressed these below and have highlighted any changes in the text of the paper in yellow.
Reviewer 3 Major points: 1. The introduction is very short. We have substantially expanded the introduction to better explain the context and prepare the reader for the proposed study. (lines 34-58)
It is hard to follow the results section. Please try to explain the results in detail for the readers to follow the content easily. We thank the reviewer for providing the following areas where we have been able to clarify aspects of the results section.
- now lines 65-67 “Analysis of IGFBP-3 in this panel of cell lines revealed a similar level of abundance” When comparing the expression levels of Hs578T and IGFBP-3, there is a major difference in the expression levels in different cell lines. IGFBP-3 is completely absent in MCF-7, T47D and MCF-10A. The authors has to try to explain the observation from the figures. We have amended this sentence to better explain the data: ‘Analysis of IGFBP-3 in this panel of cell lines revealed the highest levels of IGFBP-3 in the Hs578T cells, lower levels observed in MDA-MB-231 cells, with no detection of IGFBP-3 in the MCF-10A, MCF-7 and T47D cells (now lines 65-67).
- Better to show the immunohistochemistry data after the Western blot data, which will allow the readers to easily follow the content. We thank the reviewer for this comment. In figure 1, we have moved panel [D] to [C] and changed [C] to [D]. The text (69-71) and figure legend have been amended accordingly.
- The figure 3B i and ii is confusing. What is “csGRP78”. Explain clearly in the text.
To improve clarity, we have amended the sentence on now line 119-121 to: In order to assess changes in the abundance of cell surface (cs) proteins, we undertook biotinylation experiments which indicated that exogenous addition of IGFBP-3 increased the abundance of and translocation of GRP78 to the cell surface. We also defined cs in the figure legend.
- Figure 4 legends are wrong “(A) Representative images of GRP78 and IGFBP-3 IHC staining. (B) Kaplan-Meier plots of disease-free survival for low and high staining of IGFBP-3 in GRP78-negative tumors “. But Figure 4 indicates (A) Kaplan-Meier plots and (B) IHC staining.
Thank you for noticing this error- we have corrected the text in the legend of figure 4 accordingly.
- Please include all the supplementary figures in the result section.
- Include all the experiment results of MDA-MB-231. We have added all the data for the MDA-MB-231 cell line, that was originally in supplementary data, to figures 2 (2C) (now lines 97-98) and 3 (3B) (now lines 118-119). The text and figures legends have been amended to reflect these changes.
- Any in vivo data is available to support the in vitro data? We have not yet initiated any in vivo work.
Minor points
- (now lines 62-63): “Among the five cell lines, GRP78 staining was strongest in the highly invasive and metastatic ERα-negative Hs578T and MDA-MB-231 cells” The figure 1A i & ii indicates the protein expression using a Western blot experiment. Please change the sentence to “Among the five cell lines, GRP78 expression level was strongest in the highly invasive and metastatic ERα-negative Hs578T and MDA-MB-231 cells”
This sentence has been changed to the sentence indicated by the reviewer above.
- Now line 219 “Our results that the function of IGFBP-3 is either 207 pro- or anti-tumorigenic …. Correct the sentence.
Amended accordingly: Our results indicating that the function of IGFBP-3 is either pro- or anti-tumorigenic….

Reviewer 4 Report
This article has deciphered the association between GRP78 and IGFBP-3 and its role in tumorigenesis. The fact that IGFBP-3 exerts different effects on tumor phenotype depending on the expression of GRP78 is very interesting and novel. I think this article should be accepted, however I have a few minor questions.
- As authors have shown from Spearman’s correlation, the abundance of GRP78 and IGFBP-3 showed a positive correlation, which means very few patients have positive IGFBP-3 while expressing low GRP78. Because high IGFBP-3 can serve as a therapeutic target only in the absence of GRP78, I think targeting IGFBP-3 would not have as much clinical significance as the authors have suggested. What are authors’ thoughts on this?
- The GRP78 expression level varied among cell lines, but not precisely showed a gradual trend because MCF10A showed higher expression than cancerous cell lines(MCF7, T47D). How do the authors interpret this result?
- What does TN in table 1 stand for?
Author Response
We thank the reviewer for taking the time to read our manuscript and for making such constructive comments. We hope we have addressed these below and have highlighted any changes in the text of the paper in yellow.
Reviewer 4
- As authors have shown from Spearman’s correlation, the abundance of GRP78 and IGFBP-3 showed a positive correlation, which means very few patients have positive IGFBP-3 while expressing low GRP78. Because high IGFBP-3 can serve as a therapeutic target only in the absence of GRP78, I think targeting IGFBP-3 would not have as much clinical significance as the authors have suggested. What are authors’ thoughts on this?
We agree with the reviewer that the levels of GRP78 in MCF10A cells reflect those of the HS578T and MDA-MB-231 cells. The MCF-10A cells are a commonly used cell line for studying normal breast cell function and transformation in vitro. However, they are known to exhibit some basal-like characteristics, and therefore often express levels of some markers, such as GRP78, that are comparable to more mesenchymal than epithelial cancer cell lines.
- What does TN in table 1 stand for?
TN stands for triple negative breast cancers and has been defined in the legend for table 1 (line 85).
